# Attraction of *Aulacophora foveicollis* Lucas (Coleoptera: Chrysomelidae) to Host Plant *Cucurbita maxima* Duchesne (Cucurbitaceae) Volatiles

**Biswanath Bhowmik** [1,†,‡], **Udipta Chakraborti** [1,†,§], **Alivia Mandal** [1], **Bishwajeet Paul** [2] **and Kakali Bhadra** [1,*]

1   Department of Zoology, Kalyani University, Kalyani 741235, India
2   Division of Entomology, ICAR-Indian Agricultural Research Institute, Pusa Campus, New Delhi 110012, India
*   Correspondence: kakali_bhadra2004@yahoo.com; Tel.: +033-2582-8750 or +033-2582-8220; Fax: +033-2582-8477
†   These authors contributed equally to this work.
‡   Presently Assistant Professor from Department of Zoology, Sree Chaitanya College, North 24 Parganas, Habra 743268, India.
§   Presently in Department of Biological Sciences, Indian Institute of Science Education and Research (IISER)-Kolkata, Kalyani 741235, India.

**Abstract:** The volatiles extracted by the dynamic headspace collection system from the undamaged and conspecific damaged *Cucurbita maxima* were analyzed by Gas chromatography–mass spectrometry (GC-MS). The olfactory responses of antennal chemosensilla by male and female *A. foveicollis* towards the plant volatiles were studied by electroantennography (EAG), while the behavioral responses were analyzed by olfactometer bioassay under laboratory conditions. Scanning electron microscopic study revealed the predominance of antennal olfactory sensilla of seven different types with four types of mechanosensilla. The antennae are sexually dimorphic, with differences in density of the chemosensillae present in the apical band region of segment IX, called the circumferential band, being higher in the females. Female antennae showed maximum peak amplitudes for 2-methyl phenol (at 10 mg/mL), followed by 1,4, dimethoxybenzene (at 5 mg/mL), while male antennae showed maximum amplitudes for heneicosane (at 5 mg/mL). Y-tube bioassays revealed maximum attractiveness towards 1,4, dimethoxy benzene that decreased progressively across heneicosane, pentacosane, tetradecane, ethyl benzene, D-limonene, nonadecane, eicosane, nonanal, decanal, α-pinene, phytol and benzaldehyde in females. However, male species were more responsive towards heneicosane, followed by 1,4 dimethoxybenzene, while the responses to pentacosane and tetradecane were equal, followed by equal responses to decanal, ethyl benzene and nonadecane, and thereafter, a progressively reducing response was observed towards α-pinene, eicosane, nonanal, D-limonene, phytol and benzaldehyde. The study assists in understanding the role of olfaction by *A. foveicollis* in the host plant *Cucurbita maxima* by listing compounds that act as potential kairomones for the beetle, and can be expected to facilitate development of an eco–friendly trap and/or by attracting the natural enemies for control of the pest.

**Keywords:** electroantennography (EAG); kairomones; scanning electron microscopy SEM; straight chain hydrocarbons; volatile organic compounds VOCs; Y-tube bioassay

## 1. Introduction

The red pumpkin beetle, *Aulacophora foveicollis* Lucas (Coleoptera: Chrysomelidae), has long been a serious and major pest of cucurbitaceous plants [1–5]. It has been reported across the continents of Asia, parts of Europe and Africa [1–11]. Neonate larvae of this pest feed on healthy and young roots of the plants and feed through all the instars for approximately 12–13 days before pupation in the soil [1,2,5,7]. Later, emerged adults voraciously consume the leaf and flower tissue for about 8–9 weeks until death [1,2,5,7].

The pest does not kill the crop plant, but heavy infestation by these insects results in the death of shoots and branches, thereby causing a reduction of crop yield [12–15]. Further, they can survive in a wide range of temperature and humidity, and within the same growing season, often attack more than one crop [13–15]. This makes the control of this pest even more problematic. However, there are reports of a few natural enemies, such as mites, tachinid flies and reduviid bugs, which can help control the pests [16,17]. Nevertheless, to manage the outbreaks of this insect pest, mass releases of these biocontrol agents are needed, and this program is not yet productive. Hence, cultivators are often enforced to treat chemical insecticides indiscriminately [18,19]. To reduce yield losses and environmental risks due to insecticide application, it is very essential to develop new environment friendly sustainable products that might be included in future integrated pest management (IPM) schemes for this pest [20,21].

Investigation evidence indicates that allelochemicals involved in host location, recognition and identification may be helpful in pest management programs as baited traps [22]. However, to date, no information is available on the allelochemicals from any parts of the pumpkin plant (*Cucurbita maxima* Duchesne, Cucurbitaceae) that may act as an attractant to *A. foveicollis*. Many compounds, such as alkanes, monoterpenes, sesquiterpenes, phenyl propanoids, fatty acids and alcohols, often serve as olfactory cues for the insects to find their host [23,24]. These allelochemicals have been reported to attract [25–28] and stimulate insect oviposition [29–31]. Many more have been reported to act as an attractant to solitary bees [25], helping the insects to find their host [32].

Hence, it is of considerable interest to study the chemical profile of *Cucurbita maxima* in order to determine what contributes to host recognition by insect pests. Here, we studied the role of different synthetic compounds of various chemical classes, as well as the volatile extract emitted from *C. maxima,* that act as an olfactory cue as potential kairomones for the chrysomelid beetle. We elucidated the olfactory responses of *A. foveicollis* towards the headspace plant volatiles collected from *C. maxima* to show the preference of specific plant compounds of interest by the pest. This study specifies that semiochemicals involved in host location may contribute to novel and sustainable pest management programs, such as baited traps, in the future. The morphology of the peripheral antennal chemosensillae, olfactory stimulation, sensitivity and preference in both the sexes of *A. foveicollis* to a broad range of compounds (both natural and synthetic) of various chemical classes, by applying scanning electron microscopy SEM, electroantennogram EAG technique and olfactometer bioassay, respectively, have been investigated.

## 2. Materials and Methods

### 2.1. Insect Collection

Adults of *A. foveicollis* were obtained from several fields in North 24 Parganas of Bengal (Putia-Lat: 22.877, Long: 88.644; Kazla-Lat: 22.875, Long: 88.647; Jhikra-Lat: 22.890, Long: 88.651; Chapra-Lat: 22.873, Long: 88.621 and Pumlia-Lat: 22.884, Long: 88.593) during February–May for three consecutive years, from 2016–2019. They were reared on leaves of *C. maxima* (F1 hybrid 406 special) in the laboratory. For rearing in the laboratory, the adult pest was collected during the morning between 9–10 am from the plant (Figure S1). For the EAG experiment, the insect specimens were prepared as per the protocol reported earlier by the author [33].

### 2.2. Experimental Design

For dose–response studies, the experiments were laid out in completely randomized design with four treatments and four replications. Forty 24–48 h old insects were considered for each replication. For Y tube bioassay, as well, the experiment was laid out in a completely randomized design format with five replications comprising twenty insects in each replication. The two-way ANOVA compared the mean differences between groups that were split into two independent variables (called factors). The primary purpose of a two-way ANOVA is to understand if there is an interaction between the two independent

variables on the dependent variable. Here, our independent variables are female insects, male insects and concentrations, and the dependent variable is the response of the insects.

### 2.3. Scanning Electron Microscopy (SEM)

Classification of sensilla on the distal segments was described following the nomenclature of Zacharuk (1985) [34]. Adult insects were fixed in 2.5% glutaraldehyde in 0.1M sodium cacodylate buffer at 4 °C, overnight. Specimens were washed in buffer and post fixed in 0.1% osmium tetroxide in 0.1M sodium cacodylate buffer. The specimens were washed again in the same buffer, dehydrated in various grade of ethanols (30%, 50%, 70%, 90% and 99% for 15 min each), air-dried, placed in a vacuum desiccator (Tarson make with 300 mm diameter) for 48 h and then mounted on aluminium stubs using double-sided sticky carbon tape and coated using a sputter with gold-palladium (10–15 nm thickness). A FEI 250 (Eindhoven, The Netherlands) stereo-scan scanning electron microscope operated at 10–15 kV was used to observe the specimens.

### 2.4. Collection of Plant Volatiles from C. maxima

The plant volatile/extracts were collected from pumpkins grown on the campus, both under undamaged and herbivore-induced damaged conditions. Matured pumpkin leaves and flowers of about 20–25 gm each were used for the collection of plant volatile/extracts. The plants used for sample collection were 100–110 days old. They were collected early in the morning from 06:00 a.m. onwards till 07:00 a.m. Thereafter, the samples were placed in the refrigerator as quickly as possible to avoid loss of volatiles. Four replicates (each replicate comprising 25 leaves and flowers each, packed individually) were considered. The extracts from the undamaged and conspecific damaged plants were collected by dynamic headspace (push–pull) collection system. Charcoal filtered air was pushed into the glass jar (headspace sampling) with a constant flow rate (200 mL/min) through one inlet opening. From another opening, air laden with volatiles was pulled out with same vacuum pressure, which went through a volatile trap (Porapak Q) with the help of a vacuum pump. Uniform airflow was created inside the containers. A 6″ Porapak Q glass column (Sigma, product No. A049754) washed with HPLC grade hexane was dried and then taken for trapping the VOCs (volatile organic compounds) from the plants. The trap is essentially a synthetic polymer that is used universally. Undamaged and damaged volatiles from *C. maxima* were obtained after 48 h of extraction. The eluent was collected after washing with hexane in 2 mL vials and stored at −80 °C until use. Samples are thus ready for Gas chromatography–mass spectrometry (GC-MS) analysis.

### 2.5. Gas Chromatography-Mass Spectroscopy (GC-MS) Analysis

The separation and identification of compounds from plant extracts (column elute) were done using a Shimadzu QP 2000, Japan fitted with a Rtx-5 ms column measuring 30 mm × 0.25 mm, packed with 95% dimethyl polysiloxane, which was used for GC-MS analysis. Helium with a flow rate of 1 mL/min was used as the carrier gas with a split ratio of 1:50. One μL volume of each sample was injected into the injection port, with temperature maintained at 230 °C. The initial oven temperature was programmed at 40 °C for 4 min, and then it was increased to 220 °C with a 5 °C ramping rate and a holding time of 2 min. Finally, the temperature was increased to 270 °C with a ramping rate of 8 °C for 1 min. The running time of the sample was 60 min. The temperature for the ion source was maintained at 200 °C. Electron impact ionization (EII) with 70 eV was used for GC-MS analysis and data was evaluated by Total ion count (TIC) for the identification and quantification of compounds. The experimental protocol was followed as per earlier reports [35,36]. The mass spectra were analyzed and identified using software Turbo-Mass-OCPTVS-Demo SPL, GC-MS library NIST 14 and compared with authentic compounds and published data. The relative percentage of the identified constituents was calculated from the GC peak areas. Kovat's index was calculated for the compounds, using the retention

times of a homologous series of n-alkanes and by matching with the values reported in the literature [37–39].

### 2.6. Chemicals as Odor Stimulant

Table S1 presents a list of the volatile compounds, their purity and commercial suppliers. The compounds were selected on the basis of their widespread distribution in plants, and a few compounds were specifically selected based on their presence in *C. maxima* headspace volatile extract. Paraffin oil (Hi-Media) was used to dissolve the chemicals. Differences in volatility between the test compounds were adjusted by weighing compounds considering their specific gravity.

### 2.7. Electroantennography (EAG)

EAG responses were recorded from adults (laboratory reared, 2 days old) of *A. foveicollis,* as described earlier [33,40]. We used 80 insects (40 males and 40 females) for the entire experiment, using one antenna at a time for 35 min for~12 compounds (a time span of 80 s was given in between each stimulus). An electrode connected to a head stage pre-amplifier (Syntech) with a high impedance (>1012) to EAG amplifier (AM-02, Syntech, Hilversum, The Netherlands) was used for the recording. A signal acquisition interface board (IDAC-02, Syntech, Hilversum, The Netherlands) processed and digitized the amplified signals. The total amplification was 10X. Customized EAG software (version 2.6c, 1998, Syntech, Hilversum) computed the EAG amplitude.

### 2.8. Odor Delivery

Air and odorant were delivered with a constant flow rate of 1.8 mL/min and 0.6 L/min, respectively, for 5 s with the help of an air stimulus controller (CS-05, Syntech, Hilversum). A hexane-washed strip of filter paper (5 mm × 30 mm, Whatman No. 1) containing the test stimulus was inserted into the Pasteur pipette placed into the side port (3 mm diameter) of a tube of glass (10 mm) which served as the delivery tube, placed 20 mm from the tip of the antennae. The following concentrations of (*w/v*) were evaluated for dose-response studies: 0.1 mg, 1 mg, 5 mg and 10 mg in 1 mL of paraffin oil. Ten microlitres of each compound were pipetted onto filter paper strips and inserted into a Pasteur pipette. A pipette containing 10 μL of paraffin oil on filter paper served as the control. 1-Hexanol was used as the standard for recording the EAG responses. Responses from five antennae from different beetles were recorded per tested compound.

### 2.9. Calculation of EAG Data

Three consecutive puffs (1,2,3), along with the standard (S)1-hexanol, at each specific concentration of the odorant were taken [33,40]. The EAG amplitude spectra were obtained in millivolts (mV). These data helped in calculating the relative mean amplitudes in %, mean values, standard deviation and standard error.

### 2.10. Y-Tube Bioassay

Volatile compounds (0.1 mg/mL of each) procured from the commercial suppliers were used for olfactory bioassay for assessing the attractiveness in adult *A. foveicollis*. The test insects were kept under starvation for two–four hours before conducting the experiment to ensure a better response. The behavioral responses of adult male and female *A. foveicollis* to the volatiles were investigated in a wide mouth Y-shaped glass tube olfactometer (2.5 cm radius, 45° Y angle, 16 cm long arms). The stem of the olfactometer was connected to a porous glass vial (2.5 cm radius × 3 cm long), into which, test insects were released. Each arm of the olfactometer was connected to a glass-made micro kit adapter fitted into a glass vial (2.5 cm radius × 3 cm long). One glass vial contained a piece (1 cm$^2$) of Whatman No. 41 filter paper, moistened with 25 μL of volatiles, while the other glass vial contained a filter paper of same size, moistened with 25 μL of the control solvent (hexane). Charcoal-filtered air was pushed into the system at 300 mL min$^{-1}$. All of the connections between the

different parts of the set-up consisted of silicon tubing. A total of five replications (twenty insects per replication) for each compound, were taken. Thus, each treatment consisted of a response from 100 insects. The mean value of the responses of each compound (0.1 mg/mL) is an average of 100 insects carried out in sets of 20 in each group as one replication. The response is presented as the number of insects that responded to treatment, the number of insects that moved to the control arm and the number of insects that did not show any response/confused (i.e., those that remained undecided). The bioassays were conducted under controlled conditions in the laboratory, where temperature at 27 ± 2 °C, relative humidity RH 60 ± 5% and 165 LUX light intensity, was maintained. All the data were transformed using square root transformation before analysis.

### 2.11. Statistical Analysis

EAG values (mV) were corrected by subtracting them from the values of paraffin oil (used for diluting and making various concentrations) obtained. The data were then standardized by expressing the corrected mean EAG values (mV) as a percentage of the standard stimulus. The data recorded in percentage were then subjected to arcsine transformations. The arcsine transformation used is a standard procedure to make highly skewed distributions less skewed. This is valuable for making patterns in the data meet the assumptions of inferential statistics. These arcsines-transformed relative values were subjected to 2-way ANOVA using software SPSS 16.0. Subsequently, within each sex, a one-way ANOVA was computed. After ANOVA, post hoc analysis was performed to explore the mean differences between pairs of groups. The contrasts between chemicals were examined by the Scheffe's contrast method. *t*-test analysis was performed using PAST software version 4.05 [41] to understand the variation in the distribution of sensilla on the circumferential band of the insect pest between males and females, and along with this, *t*-test analysis was also used for visualizing the comparative aspects between treatment and control groups in olfactory bioassay. Homogeneity of variance was tested for all the data by performing Levene's test (untransformed data), in which, non-significant values were assumed to be homogenous (equal variance) and homogenous data were subjected to the *t*-test equal variance analysis.

## 3. Results

### 3.1. Collection and Analysis of C. maxima Plant Volatiles

GC-MS spectrum analysis of plant volatiles emitted from *Cucurbita maxima*, both undamaged and conspecific damaged, confirmed the presence of various emitted volatile organic compounds (VOCs) that led to the identification of a number of bio-active compounds of the plant with different retention times, as illustrated in Figure 1A,B. A total of 22 peaks were identified in the chromatogram from undamaged *C. maxima* (Figure 1A). The major volatile substances, which were prominent and dominant (based on peak area) in the tested parts of the undamaged (Table 1) plant, showed a list of several long-chain saturated hydrocarbons/n-alkane, few aromatic compounds and terpenoids. Among the volatile substances of the plant, 1, 4 dimethoxybenzene showed the highest peak area of 12.25%, followed by heneicosane (7.77%), pentacosane (7.11%), tetradecane (6.51%), nonadecane (6.13%) and D-limonene (6.05%). Apart from these, the extract also contains a considerable amount of α-pinene (5.95%), decanal (5.94%), benzaldehyde (5.61%), nonanal (5.57%), ethylbenzene (4.76%) and eicosane (4.20%). Additionally, heptadecane (3.52%), phytol (3.41%), furan 3-4 methyl-3 pentenyl (2.90%), eucalyptol (2.55%), hexadecane (2.37%), pentadecane (2.31%), octadecane (1.77%), 4-methyl phenol (1.04%), 2-ethyl 1-hexanol (0.891%) and tetradecane 2-6-10 trimethyl (0.878%) were obtained in traces. In a similar study, the volatile organic compounds emitted from conspecific damaged plants result in an increase in the emission of volatile compounds such as phenol 4-methyl, 1,4 dimethoxybenzene, decanal, hexadecane, heptadecane, octadecane, eicosane, heneicosane and pentacosane (Figure 1B).

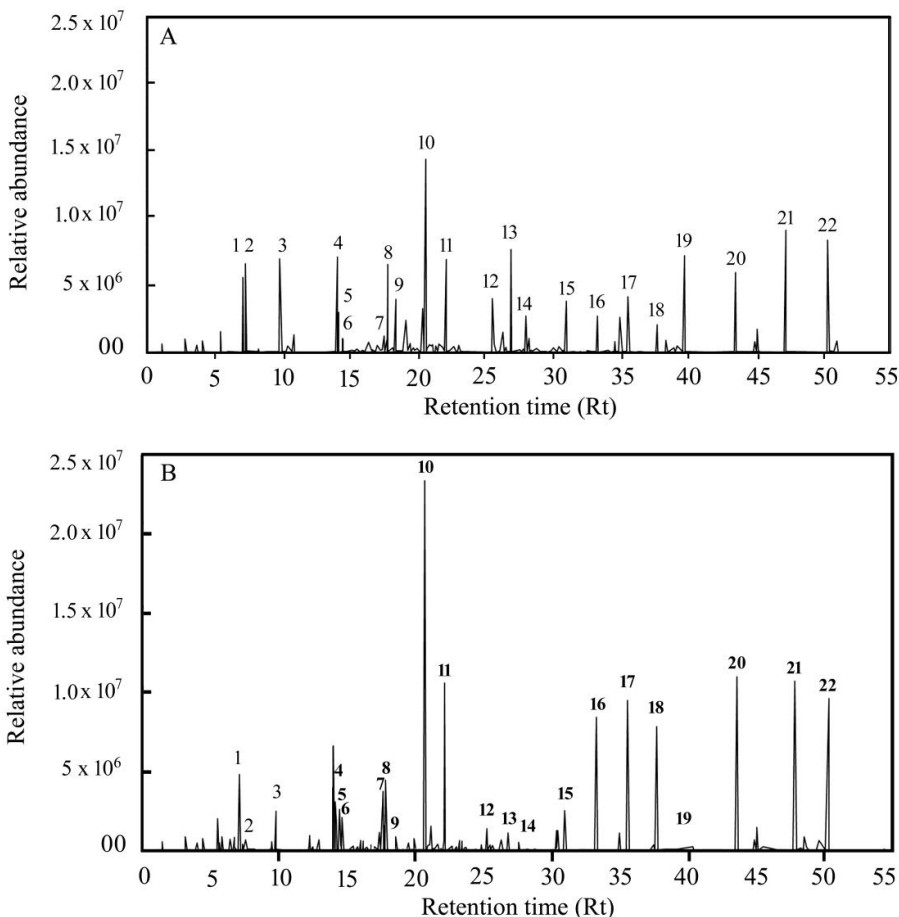

**Figure 1.** (**A**) Chromatographic profiles of volatiles from the undamaged *Cucurbita maxima* host plant of *Aulacophora foveicollis*. The numerical values above the peak denote the peak numbers (Table 1). (**B**) Chromatographic profiles of volatile extracts emitted from the conspecific damaged *Cucurbita maxima* host plant.

**Table 1.** Volatiles extracted by headspace from undamaged *Cucurbita maxima*.

| Peak No. | RT | Area Peak | Library/ID | CAS No. |
|---|---|---|---|---|
| 1 | 7.1005 | 5.5762 | Ethylbenzene (aromatic) | 000100-41-4 |
| 2 | 7.2904 | 6.5621 | Benzaldehyde (aromatic) | 000100-52-7 |
| 3 | 9.7898 | 6.9695 | α-Pinene (terpenoid) | 007785-70-8 |
| 4 | 14.0769 | 7.0872 | D-Limonene (terpenoid) | 005989-27-5 |
| 5 | 14.1797 | 2.989 | Eucalyptol (terpenoid) | 000470-82-6 |
| 6 | 14.4328 | 1.0426 | 1-Hexanol, 2-ethyl- | 000104-76-7 |
| 7 | 17.5492 | 1.217 | Phenol, 4-methyl- | 000106-44-5 |
| 8 | 17.7944 | 6.5213 | Nonanal (C9) | 000124-19-6 |
| 9 | 18.3956 | 3.9556 | Furan, 3-(4-methyl-3-pentenyl)- | 000539-52-6 |
| 10 | 20.6103 | 14.3379 | 1,4-dimethoxybenzene (aromatic) | 000150-78-7 |
| 11 | 22.1211 | 6.9088 | Decanal (C10) | 000112-31-2 |
| 12 | 25.546 | 3.9962 | Phytol (terpenoid) | 000150-86-7 |
| 13 | 26.8985 | 7.6281 | Tetradecane (C14) | 000629-59-4 |
| 14 | 27.9979 | 2.7089 | Pentadecane (C15) | 0629-62-9 |
| 15 | 28.2273 | 1.0281 | Tetradecane, 2,6,10-trimethyl- | 014905-56-7 |
| 16 | 33.2895 | 2.7779 | Hexadecane (C16) | 000544-76-3 |
| 17 | 35.5438 | 4.126 | Heptadecane (C17) | 000629-78-7 |
| 18 | 37.6874 | 2.0709 | Octadecane (C18) | 000593-45-3 |
| 19 | 39.7201 | 7.1811 | Nonadecane (C19) | 0629-925-5 |
| 20 | 43.5089 | 4.9217 | Eicosane (C20) | 013475-77-9 |
| 21 | 47.2321 | 9.0922 | Heneicosane (C21) | 000629-94-7 |
| 22 | 50.3112 | 8.3289 | Pentacosane (C25) | 000629-99-2 |

Note: Area peaks below 1.0281 were not considered.

### 3.2. Distribution and Variation of Antennal Chemosensilla in A. foveicollis

The antennae of adult *A. foveicollis* consist of a scape, a short pedicel and a flagellum, bearing nine segments (Figure 2A,B, upper panel). There was not much difference in the morphology of the sensilla between male and female specimens, rather, the difference lies in the distribution (density) of the sensilla, especially in the apical band region of segment IX, called the circumferential band, which was lower in male insects than in female insects (Figure 2C,D, upper panel). Table 2 represents the distribution, variations and *t*-test analysis of sensilla on the dorsal and ventral side of the circumferential band in the antenna. From SEM images (Figure 2A–I, Lower panel), broadly, three types of thick, short and long sensilla with longitudinal grooves have been observed. Their distribution is mostly on the lateral surfaces.

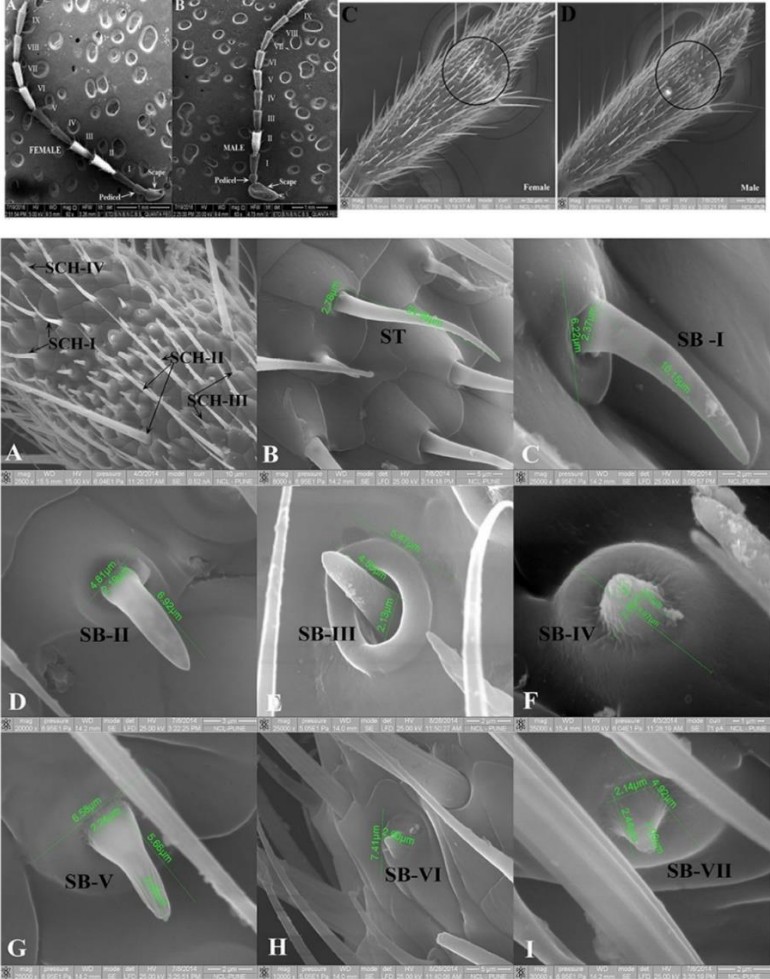

**Figure 2.** (Upper panel) (**A,B**) Antennae of adult *Aulacophora foveicollis*. (**C,D**) Location of circumferential band in the apical region of IXth segment in female and male *A. foveicollis*. (Magnification 63×–700×). (Lower panel) (**A–I**) Variations and distributions of antennal chemosensilla in *A. foveicollis*. Note: SB: Sensilla Basiconica; SCH: Sensilla Chaetica; ST: Sensilla Trichodea (Magnification 2500×–35,000×).

### 3.2.1. Sensilla Chaetica

Four types of sensilla chaetica are present on the apical segment. Type I is short with a slightly curved apical pointed tip, Type II is elongated with a pointed tip, Type III has a short and pointed tip and Type IV is morphologically similar to Type II and III but is of moderate size. They all look like bristles with sharp pointed ends, and are characterized by longitudinal grooves, are aporous and vary in length from 48–105 μM. Mostly, they

are mechanosensilla in function (Figure 2 Lower panel A). *t*-test analysis results showed a significant difference in the distribution of sensilla chaetica type I (*t*-test value = 3.9, df = 18, *p* < 0.05) and type II (*t*-test = 2.7, df = 18, *p* < 0.05) among males and females from the dorsal side (Table 2).

**Table 2.** Distribution of sensilla on the circumferential band among male and female *Aulacophora foveicollis*.

| | SB-I | SB-II | SB-III | SB-IV | SB-V | SB-VI | SB-VII | SCH-I | SCH-II | ST |
|---|---|---|---|---|---|---|---|---|---|---|
| **Dorsal Side (mean ± se)** | | | | | | | | | | |
| **Male** | 7.1 ± 0.4 | 5.4 ± 0.3 | 3.6 ± 0.3 | 3.5 ± 0.4 | 5.3 ± 0.5 | 4.8 ± 0.2 | 4.6 ± 0.3 | 5 ± 0.3 | 5.6 ± 0.3 | 5.2 ± 0.4 |
| **Female** | 6.8 ± 0.4 | 6 ± 0.3 | 6.3 ± 0.5 | 5.3 ± 0.5 | 5 ± 0.4 | 7.1 ± 0.4 | 6.5 ± 0.4 | 7.3 ± 0.4 | 7.3 ± 0.4 | 3 ± 0.3 |
| *t*-test value | 0.47 | 1.1 | **4.2 \*** | **2.5 \*** | 0.39 | **4.2 \*** | **3.2 \*** | **3.9 \*** | **2.7 \*** | **3.8 \*** |
| *p*-value | 0.64 | 0.28 | 0.0004 | 0.01 | 0.69 | 0.0004 | 0.004 | 0.0008 | 0.01 | 0.001 |
| **Ventral side (mean ± se)** | | | | | | | | | | |
| **Male** | 3.5 ± 0.2 | 5.4 ± 0.5 | 3.5 ± 0.4 | 3.5 ± 0.4 | 3.8 ± 0.3 | 3.6 ± 0.5 | 4 ± 0.3 | 6 ± 0.4 | 4.9 ± 0.4 | 3.4 ± 0.4 |
| **Female** | 6.2 ± 0.3 | 5.8 ± 0.4 | 5.3 ± 0.5 | 3.9 ± 0.4 | 3.9 ± 0.4 | 4.8 ± 0.3 | 6.1 ± 0.4 | 6.9 ± 0.6 | 5.4 ± 0.4 | 3.7 ± 0.3 |
| *t*-test value | **6.3 \*** | 0.58 | **2.4 \*** | 0.62 | 0.17 | 1.8 | **3.4 \*** | 1.1 | 1.78 | 0.57 |
| *p*-value | *p* < 0.001 | 0.56 | 0.02 | 0.54 | 0.86 | 0.08 | 0.002 | 0.26 | 0.11 | 0.57 |

**Abbreviations:** SB: SensillaBasiconica; SCH: Sensilla Chaetica; ST: Sensilla Trichodea. An average of five insect individuals each from male and female, respectively, were taken for the study (N = 10). **Bold *t*-test values** with "*" are subjected to be significant at *p* < 0.05.

### 3.2.2. Sensilla Trichodea

About 24.99 μM × 2.78 μM in dimensions, uniporous with elongated sensillar shaft. Distributed uniformly across the apical segment. Functioning as chemosensillae (Figure 2B, Lower panel). The distribution of the sensilla trichodea varied significantly among the both sexes from the dorsal side only (*t*-test = 3.8, df = 18, *p* < 0.05) (Table 2).

### 3.2.3. Sensilla Basiconica

Seven types were observed on the apical segment, involved in olfaction. Type I: about 10.15 μM × 2.37 μM in dimensions with a short shaft (Figure 2C, lower panel). Type II: about 6.92 μM × 2.19 μM in dimensions (Figure 2D, Lower panel) with a similar short shaft. Type III: about 4.55 μM × 2.13 μM in length–breadth, and the base of the sensillar shaft forms a depression around the entire circumference within the socket (Figure 2E, Lower panel). Type IV: very short sensilla (4.97 μM × 2.47 μM) with an apical pore, having five papillae that extend up to 1 μM (Figure 2F, Lower panel). Type V: characterized by apical finger like projections (5.66 μM × 2.24 μM) (Figure 1 Lower panel G). Type VI: very short cone like structure (7.41 μM × 2.90 μM) with the apical tip barely reaching above the level of the socket (Figure 2H, Lower panel). Type VII: similar to type IV but without papillae, with 2.44 μM × 2.14 μM dimensions (Figure 2I, Lower panel). The distribution of sensilla basiconica type III, type IV, type VI and type VII varied significantly among males and females for the dorsal side only, and for the ventral side, type I, type III and type VII varied significantly. While in the case of sensilla basiconica type II and type V, there was no significant variation found in their distribution on the circumferential band among males and females (Table 2).

### 3.3. EAG Dose–Response Studies in A. foveicollis

Both males and females of *A. foveicollis* responded to 50 volatile plant substances of various concentrations (0.1, 1.0, 5.0 and 10.0 mg/mL) and chemical classes evaluated from the EAG studies, with differences in response. Compounds were selected based on their presence in a considerable amount in the volatile extract of *C. maxima* and as common plant volatile. *A. foveicollis* is a polyphagous pest in the field condition, where it comes across plethora of compounds that are available in the field and hence other common plants volatiles in different doses were also selected for further EAG analysis. Figure S2 illustrates the raw data of representative EAG spectra measured in mV of few stimulated odorants at a specific concentration along with its standard 1-hexanol. The mean response of *A. foveicollis* males at 10.0, 5.0, 1.0 and 0.1 mg/mL stimulation by 1-Hexanol (standard) was 2.938 ± 0.26, 3.031 ± 0.32, 1.616 ± 0.16 and 1.885 ± 0.48 mV, respectively. While, in

females, it was 2.775 ± 0.64, 2.066 ± 0.12, 1.546 ± 0.34 and 1.828 ± 0.52 mV at 10.0, 5.0, 1.0 and 0.1 mg/mL, respectively. Total 50 compounds, procured from different chemical agencies (Table S1), were tested for their EAG dose dependent responses in female (Table S2) and male insects (Table S3).

### 3.3.1. Aliphatic Compounds

Among the aliphatic compounds, females with Trans-2-Hexen-1-ol elicited the significantly highest EAG amplitude at 5.0 mg/mL (119.2 ± 1.87%), followed by 90.48 ± 1.16% at 1 mg/mL, 86.62 ± 2.47% at 10 mg/mL and 65.68 ± 0.84% at 0.1 mg/mL. Alternately, straight chain alkanes like Nonadecane elicited a maximum EAG response of 96.96 ± 1.82% at 10 mg/mL, followed by 66.21 ± 1.82, 51.72 ± 2.11 and 41.29 ± 2.01% at 5, 0.1 and 1 mg/mL, respectively. Tetradecane, on the other hand, showed a maximum response of 89.31 ± 1.62% at 10 mg/mL, while other data followed as 69.0 ± 2.32%, 40.21 ± 1.11% and 31.29 ± 1.11% at 5, 10 and 0.1 mg/mL doses, respectively. Pentacosane elicited a maximum response of 95.21 ± 2.11% at 10 mg/mL, while at 0.1, 1 and 5 mg/mL, it showed 44.69 ± 2.11%, 39.72 ± 1.11% and 80.79 ± 1.78%, respectively. With heneicosane, the maximum response was observed at 10 mg/mL (100.21 ± 2.32%) and the minimum response was elicited at 1 mg/mL (39.29 ± 0.96%). Eicosane elicited a maximum response of 78.21 ± 1.21% at 10 mg/mL, while the responses at 0.1, 1 and 5 mg/mL were 41.21 ± 1.99%, 52.32 ± 1.62% and 60.7 ± 0.79%, respectively. Heptadecane, on the other hand, showed a maximum response of 65.10 ± 1.72% at 10 mg/mL, while other data followed as 39.72 ± 1.72%, 41.72 ± 1.32% and 63.79 ± 2.32% at 0.1, 1 and 5 mg/mL, respectively. Among male insects, Heneicosane showed the maximum response at 5 mg/mL (129.3 ± 1.99%), while at 0.1, 1 and 10 mg/mL of doses, the responses were 65.00 ± 1.72%, 101.21 ± 1.21% and 111.99 ± 1.81%, respectively. A minimum amplitude of 2.48 ± 1.38% was reported with acetic acid at a concentration of 10 mg/mL. Other values with acetic acid read as 19.50 ± 1.30%, 11.65 ± 3.48% and 11.59 ± 1.40% at 0.1, 1 and 5 mg/mL of doses, respectively. Figure 3A,B and Tables S2 and S3 can be consulted for the mean responses of other aliphatic compounds in female and male specimens, respectively.

### 3.3.2. Aromatic Compounds

The responses of dose-dependent aromatic compounds also elicited similar tendency of changes in EAG amplitudes among female specimens of *A. foveicollis* (Figure 3C and Table S2). 2-methyl phenol elicited the significantly highest EAG amplitude at 10.0 mg/mL (136.51 ± 0.97%), while at 0.1 mg/mL, the least amplitude close to 7.71 ± 0.75% with 4-methyl phenol was reported among female insects. Benzyl alcohol showed the highest EAG amplitude of 100.99 ± 1.83%, followed by 92.71 ± 0.98%, 74.59 ± 2.80% and 66.76 ± 2.34% at 10. 5, 0.1 and 1 mg/mL doses, respectively. Acetophenone elicited a maximum response at 10 mg/mL (65.17 ± 1.49%). Benzaldehyde showed a maximum amplitude of 55.14 ± 0.98% at 0.1 mg/mL followed by 51.06 ± 0.56%, 31.10 ± 1.00% and 22.24 ± 0.51% at 10, 1 and 5 mg/mL doses, respectively. Phenylacetaldehyde, on the other hand, elicited the highest EAG stimulation of 62.036 ± 1.79% at 5 mg/mL. Other aromatic compounds, such as Isoeugenol, Phenethyl alcohol, 2 methyl phenol, 3 methyl phenol and Ethyl benzoate showed maximum amplitudes of 40.08 ± 0.99%, 119.37 ± 1.09%, 136.51 ± 0.97%, 120.08 ± 2.26% and 62.87 ± 2.05%, respectively, at 10 mg/mL. 4-methyl phenol showed the highest response at 5 mg/mL (88.905 ± 0.73%), followed by 66.49 ± 1.59% at 10 mg/mL and 28.18 ± 0.54% at 1 mg/mL, and a minimum response of 7.71 ± 0.75% at 0.1 mg/mL. 1,4 dimethoxybenzene elicited a significantly high EAG amplitude at 5.0 mg/mL (122.4 ± 1.23%), while at 10 mg/mL, showed 119.82 ± 1.89%, followed by 95.50 ± 0.49 and 69.16 ± 1.11% at 1 and 0.1 mg/mL doses among female species, respectively. Ethyl benzene, on the other hand, showed the highest EAG amplitude of 100.51 ± 1.21% at 10 mg/mL, followed by 80.76 ± 1.11%, 61.71 ± 1.09% and 50.43 ± 1.29% at 5, 0.1 and 1 mg/mL doses, respectively.

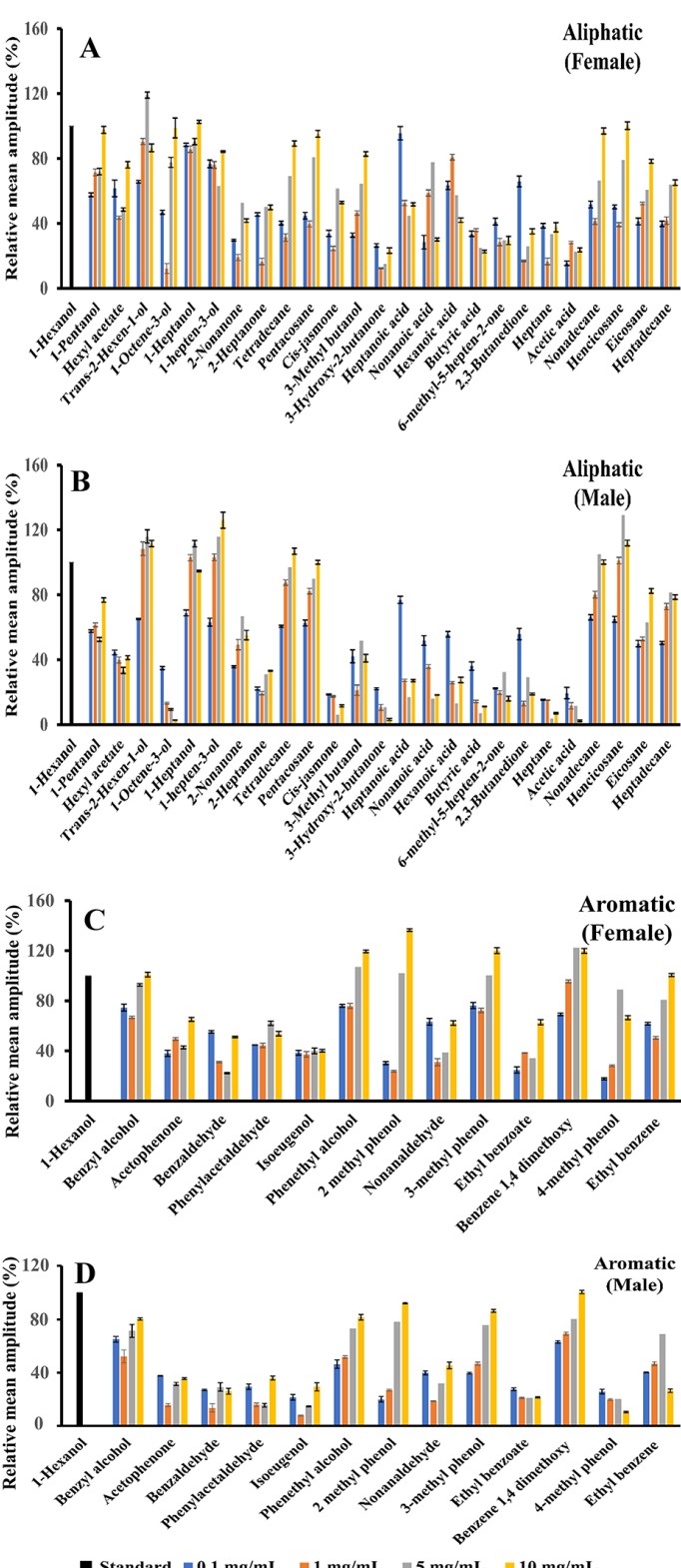

**Figure 3.** Mean normalized electroantennography (EAG) responses for aliphatic (**A**,**B**) and aromatic (**C**,**D**) compounds in female and male *Aulacophora foveicollis* to plant volatiles at 0.1, 1, 5.0 and 10 mg/mL concentration.

Males elicited maximum changes of 100.50 ± 1.25% in EAG amplitudes with 1,4, dimethoxybenzene at 10 mg/mL, while the other readings followed as 80.25 ± 1.11%, 69.14 ± 0.61% and 62.99 ± 1.02% at 5, 1 and 0.1 mg/mL concentrations, respectively.

Minimum changes of $3.71 \pm 2.42\%$ were observed with indole at 10 mg/mL. Other values can be followed according to the bar graph presentation in Figure 3D, and the values are collated in Table S3.

### 3.3.3. Green Leaf Volatiles (GLV)

Among female insects, cis-3-hexen-1-ol at a dose of 5 mg/mL elicited greater responses ($113.4 \pm 1.23\%$), while decanal showed minimum responses ($17.80 \pm 0.13\%$) at the lowest doses of 0.1 mg/mL. As with females, male insects also showed a similar trend, but the dose responses were different. Among males, cis-3-hexen-1-ol at a dose of 10 mg/mL elicited greater responses ($121.70 \pm 2.44\%$), while decanal showed minimum responses ($10.96 \pm 1.42\%$) at the lower doses of 1.0 mg/mL. Dose-dependent stimulation of other green leaf volatile compounds among female and male insects can be compared based on Figure 4A,B and Tables S2 and S3.

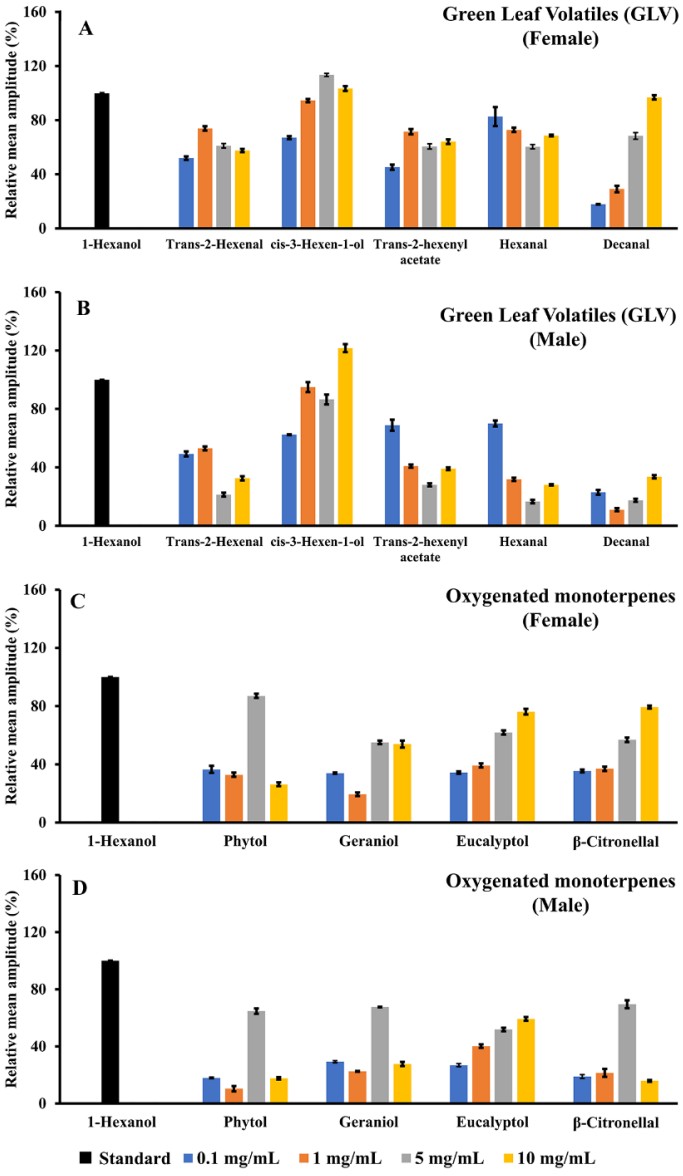

**Figure 4.** Mean normalized electroantennography (EAG) responses for GLV (**A,B**) and oxygenated monoterpenes (**C,D**) compounds in female and male *Aulacophora foveicollis* to plant volatiles at 0.1, 1, 5.0 and 10 mg/mL concentration.

### 3.3.4. Oxygenated Monoterpenes

Equally varying EAG responses were observed with oxygenated monoterpenes (Figure 4C,D; Tables S2 and S3). In females, Phytol induced maximum responses at 5.0 mg/mL (91.47 ± 2.31%) and Geraniol elicited a maximum response of 55.062 ± 1.24% at 5 mg/mL. While β-Citronellal showed maximum EAG amplitudes of 79.30 ± 1.00% at 10 mg/mL. Eucalyptol, on the other hand, showed the highest EAG amplitude of 76.13 ± 1.92% at 10 mg/mL, followed by 61.83 ± 1.41%, 39.21 ± 1.31% and 34.32 ± 0.81% at 5, 1 and 0.1 mg/mL doses, respectively.

In males, among oxygenated monoterpenes, Phytol showed the maximum and minimum responses of 86.50 ± 2.70% and 10.40 ± 2.28% at 5.0 and 1.0 mg/mL, respectively. While other compounds, such as Geraniol, elicited a maximum response of 64.66 ± 1.54% at 5 mg/mL. β-Citronellal, on the other hand, elicited maximum EAG amplitudes of 69.53 ± 2.29% at 5 mg/mL. Eucalyptol showed the highest EAG amplitude of 59.21 ± 1.32% at 10 mg/mL, followed by 51.83 ± 1.21%, 40.21 ± 0.99% and 26.84 ± 1.11% at 5, 1 and 0.1 mg/mL doses, respectively.

### 3.3.5. Sesquiterpenes

There were not many significant differences reported among females and males in response with α-humulene (Figure 5A,B, Tables S2 and S3). Among females, α-humulene at 5 mg/mL and 10 mg/mL, respectively, showed the maximum and minimum amplitudes (51.08 ± 0.38%; 12.20 ± 0.46%), males, on the other hand, showed maximum and minimum amplitudes of 51.08 ± 0.38% and 8.46 ± 1.40% at 5 mg/mL and 10 mg/mL, respectively.

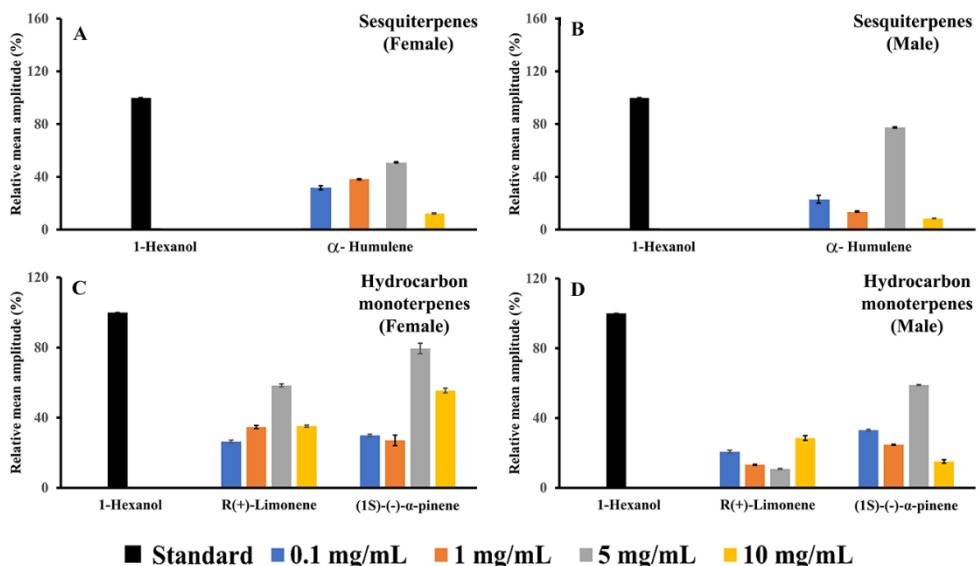

**Figure 5.** Mean normalized electroantennography (EAG) responses for sesquiterpenes (**A**,**B**) and hydrocarbon monoterpenes (**C**,**D**) compounds in female and male *Aulacophora foveicollis* to plant volatiles at 0.1, 1, 5.0 and 10 mg/mL concentration.

### 3.3.6. Hydrocarbon Monoterpenes

α-pinene in female specimens elicited the highest EAG amplitude of 79.52 ± 2.96% at 5 mg/mL, followed by 55.48 ± 1.35%, 29.89 ± 0.72% and 27.04 ± 2.12% at 10, 0.1 and 1 mg/mL doses, respectively. While α-pinene in males showed 58.94 ± 1.30, 33.01 ± 2.30, 24.69 ± 2.63 and 15.09 ± 1.90% at 5, 0.1, 1 and 10 mg/mL concentrations, respectively. D(+)-Limonene elicited maximum amplitudes of 58.39 ± 0.872% at 5 mg/mL and 28.54 ± 1.66% at 10 mg/mL in female and male *A. foveicollis*, respectively (Figure 5C,D; Tables S2 and S3).

### 3.4. Analysis by Two-Way ANOVA and Post Hoc Tests

Two-way ANOVA and post hoc tests analysis (Table S4A–H) were done for visualizing the effect of concentrations of the odorants on response by gender. The data showed a significant difference between the responses of males and females (F = 49.991, *df* = 1, *p* < 0.05; Partial Eta Squared = 0.113). Similarly, the effect of concentration (0.1, 1, 5 and 10 mg/mL) also showed a significant effect on the response (Table S4C) (F = 6.476, *df* = 3, *p* < 0.05, Partial Eta Squared = 0.047). In respect to the combining effect of gender and concentration on response, it showed a non-significant interaction effect (F = 2.029, *df* = 3, *p* = 0.109, Partial Eta Squared = 0.015).

To explore the mean differences between pairs of groups (between concentrations 0.1,1, 5 and 10 mg/mL), a post hoc Tukey HSD (Honestly significant difference) test analysis was performed (Table S4F). The respective mean differences and significant levels were calculated and presented in Table S4F. From this multiple comparison of the responses of EAG amplitudes at different concentrations, it was further observed that different groups could be set in homogeneous subsets. In homogeneous groups, there were no significant differences. Based on the table values, we can say that 1 mg/mL and 0.1 mg/mL concentrations make up the homogeneous group, while 0.1 mg/mL, 10 mg/mL & 5 mg/mL concentrations were in another homogeneous set (Table S4G). Both of the groups were significantly different from each other. The data were graphically presented, and it was shown that, though the patterns of the plots were same, a higher EAG response was shown in females than males, with increasing concentration (Table S4H).

### 3.5. Analysis of R2 Regression Values of the Transformed Data in Aulacophora foveicollis

The study further highlights and analyzes the R2 values for both male and female insects. R-squared is a goodness-of-fit measure for regression models. The regression is of the polynomial type and follows a quadratic second order equation. The mean values presented are worked out after doing square root transformation of the data. The square root transformation was done using x = $\sqrt{}$ (0.5+ observed value). Since it is a second-order polynomial equation, the fundamental theorem of algebra guarantees that it has two solutions. The slope of a quadratic function changes at each point along the function. It was found by taking the derivative of the function and evaluating the function at the point in question. Compounds such as 1-heptanol, hexanal, butyric acid, Trans-2-hexenyl acetate, 1,4, dimethoxybenzene, hexanoic acid, Eicosane, heptanoic acid, Tetradecane, Pentacosane, Phenylacetaldehyde, 1-hepten-3-ol, Decanal, lactic acid, Trans-2-Hexen-1-ol, Isoeugenol, 1-Octene-3-ol, Nonanoic acid, heneicosane, 4-methyl phenol, Phenethyl alcohol, 3-methyl phenol, R(+)-Limonene, 2 methyl phenol, Nonadecane, Acetic acid, 3-Hydroxy-2-butanone, 2-Nonanone, Nonanal and cis-3-Hexen-1-ol at the different doses have R2 values near to 1, and are listed as the most responsive compounds. Moderately responsive compounds *viz.* Hexyl acetate, Ethyl benzoate, Benzyl alcohol, 2-Heptanone, Pentyl acetate, Cis-jasmone, Acetophenone, Trans-2-Hexenal, β-Citronellol, Indole and β-Citronellal at the different doses have R2 values between <0.8–0.4. Erratic responsive compounds (Phytol, 6-methyl-5-hepten-2-one, (1S)-(-)-α-pinene, α-Humulene, Citral, Geranyl acetate, Geraniol, Fernesol, Benzaldehyde, 2-Pentanone, Undecane and 3-Methyl butanol) ar different doses have values R2 < 0.4 (individual values not shown).

### 3.6. Y-Tube Assay

Based on the headspace volatile extracts from *C. maxima* and stimulation in EAG responses, compounds were selected for the Y-tube olfactometric bioassay. Figure 6A,B shows the bar graph representation of the olfactometric data in female and male specimens of *A. foveicollis*. Overall, female pests showed more response than males. However, attractiveness towards decanal, nonadecane, α-pinene and eicosane was shown to be higher in male insectes compared to female insects. Table S5 represents the numerical values of the Y-tube assay in females and males. Females were more responsive towards 1,4 dimethoxybenzene > heneicosane (C21) > pentacosane (C25) > tetradecane (C14) > ethyl

benzene > D-limonene> nonadecane (C19)> eicosane (C20)> nonanal (C9)> decanal (C10)> α-pinene> phytol> benzaldehyde. On the other hand, male species showed more responses towards heneicosane (C21) > 1,4 dimethoxybenzene > pentacosane (C25) = tetradecane (C14) > decanal (C10) = ethyl benzene = nonadecane (C19) > α-pinene > eicosane (C20) > nonanal (C9) > D-limonene> phytol> benzaldehyde. The trend shows an overall preference of straight chain hydrocarbons in both the sexes.

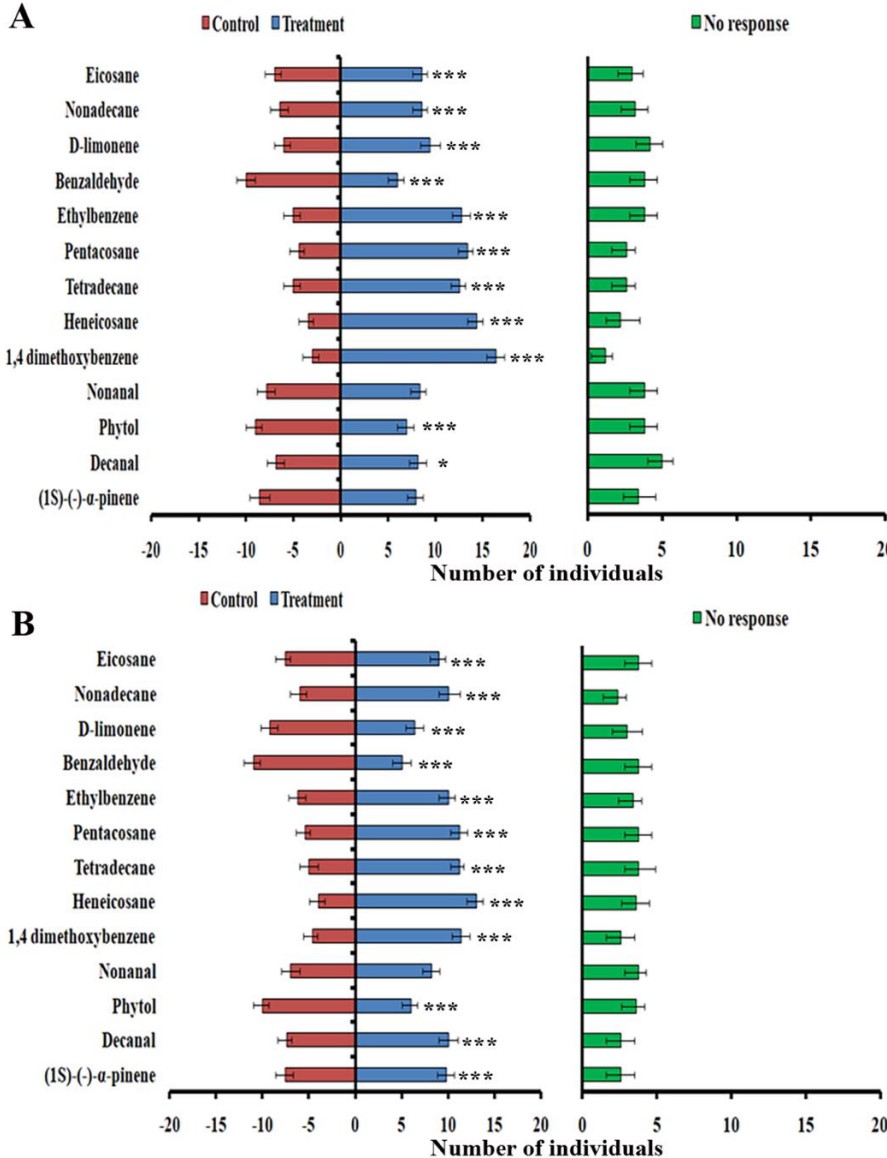

**Figure 6.** Y-tube bioassay response of female (**A**) and male (**B**) *Aulacophora foveicollis* at 0.1 mg/mL concentration. Significant differences are subjected to '*' $p < 0.05$ and '***' $p < 0.001$ respectively in *t*-test analysis between the response of treatment and control; in graph '*' is placed on the preferred side relating to the response of *Aulacophora foveicollis*. A total number of five replicates with 20 individuals in each have been studied in this assay, i.e., N = 100.

Based on *t*-test analysis, compounds, other than nonanal and α-pinene, showed significant differences in response among the treatment and control group of species in female *A. foveicollis*, while in male *A. foveicollis*, nonanal was the only compound with a non-significant difference between the treatment and control group (Table S5). These non-significant *t*-test values between the two groups showed the relatively less-attractive nature of the compounds α-pinene and nonanal.

## 4. Discussion

Among a total of 22 identified volatile organic compounds (VOCs) emitted from the *C. maxima*, 1,4 dimethoxybenzene was found in the maximum amount. It is a very significant attractant for Chrysomelidae [42]. Following this were saturated straight chain hydrocarbons/n-alkanes such as heneicosane (C21), pentacosane (C25), tetradecane (C14), nonadecane (C19), eicosane (C20) and heptadecane (C17), which were also found in measurable amounts in the list of emitted VOCs from the plant [43–45]. Alkanes are the most common compounds of cuticular waxes of plants [46,47], and they play an important role in herbivore–plant interactions as feeding attractants [23–28,48]. They are also very effective female attractants for oviposition [30,49]. Again, D-limonene, the terpenoid obtained from the list of emitted plant volatiles, is also reported to have insect feeding-attractant properties [43]. Nonanal, along with a few other volatiles, has been reported to act as an *A. foevicollis* female attractant in *Momordica cochinchinensis* [9,10]. On the other hand, the plant volatile ethyl benzene is also tested as an attractant for many insect pests [50]. Phytol is another terpenoid volatile emitted from the plant, and is reported to attract female *A. foveicollis* [9,10]. Apart from the extracted volatiles from the undamaged Cucurbita, volatiles obtained from the damaged plants are also ecologically significant. Herbivore-induced plant volatiles can provide a direct defense by being unpalatable or repelling to the phytophagous insect, or can provide an indirect defense by attracting natural enemies [50,51]. Under normal conditions, plants released small quantities of these volatiles, but when they are attacked by any herbivorous insects, the emission of volatiles from the plants are significantly increased under this stressful condition [52]. Our study also revealed a similar trend where herbivore feeding resulted in a higher amount of emitted volatile organic compounds *viz.* 1,4 dimethoxybenzene, decanal, hexadecane, heptadecane, octadecane, eicosane, heneicosane, and pentacosane. Chromatographic analysis of the pest-infested plant volatiles showed the dominance of saturated straight chain hydrocarbons. These compounds play an important role in plant defense induced by herbivory by facilitating the attraction of natural enemies to the pests [50,51,53].

Scanning electron microscopic study showed sexual dimorphism among female and male insects of *A. foveicollis*. Female specimens have a greater number of Sensilla Chaetica Type-I on both the dorsal and ventral sides of the antennal segment. Whereas males have a greater number of Sensilla Basiconica Type-I on the dorsal side, with a greater number of Sensilla Chaetica Type-I on the ventral side of the circumferential band. Overall, the density of chemosensillae present in the apical band region of the circumferential band is higher in the female species (Table 2). The study revealed the predominance of olfactory sensilla, being more present in female insects compared to male insects, resulting in greater responses and preferences in females.

Furthermore, these insects showed conspicuous sexual dimorphism based on the differences in EAG responses under laboratory conditions to the various compounds at different doses. The EAG data here reflected a broad range of feeding habitats for the insect pests. The females showed a maximum peak amplitude for 2-methyl phenol ($136.51 \pm 0.97$ at 10 mg/mL), followed by 1,4, dimethoxybenzene ($122.4 \pm 1.23$ at 5 mg/mL) and cis 3-hexene-1-ol ($113.4 \pm 1.23$ at 5 mg/mL), while males showed a maximum amplitude for heneicosane ($129.3 \pm 1.99$ at 5 mg/mL), followed by 1-hepten-3-ol ($126.11 \pm 2.82$ at 10 mg/mL) and cis 3-hexene-1-ol ($121.70 \pm 2.44$ at 10 mg/mL). The reason for the higher responses for aromatic compounds compared to aliphatic compounds in females might be the fact that these aromatic compounds are important constituents of pheromones in this insect pest. Meanwhile, the greater sensitivity towards GLV shown in females may suggest a species-specific adaptation of the set of olfactory receptors on the antennae to the particular green color components of the cucurbitaceae family. Moreover, preference for green leaf volatile compounds appeared to be an important factor for host plant selection by the pumpkin beetle, thereby suggesting its polyphagous nature [54]. The ability of both males and females to detect the same odor is probably due to the use of the same cues to locate the host plants for survival and reproduction in similar habitats. In several

species of phytophagous insects, such as the potato tuber moth [*Phthorimaea operculella* (Zeller, 1873); Family: Gelechiidae; Order: Lepidoptera] and apple maggot [*Rhagoletis pomonella* (Walsh,1867); Family: Tephritidae; Order: Diptera], it has been documented that the choice of a particular host plant depends on the bouquet of compounds released in a particular ratio [54–56]. Variations observed in dose–response studies may be due to differences in the release rates of different compounds [57] and to the sensitivity of the olfactory receptor system, thereby reflecting differential tuning of receptors [58]. However, the significance of higher responses in females than males towards specific plant volatiles in this study is still not fully explained at this point. It is possible that stronger stimuli may induce a general increase in responsiveness, and thus, preferences may be altered and temporary specialization to preferred stimuli may arise, in view of changes in concentration of the volatile components. Among terpenoids, sesquiterpenes elicited a significantly lower EAG response compared to hydrocarbon monoterpenes and oxygenated monoterpenes. Terpenoids at 0.1, 1.0 and 10.0 mg/mL elicited lower responses in both male and female species. Comparatively, at 5.0 mg/mL, the response was higher for phytol, α-pinene and α-humulene, signifying the optimum concentration for the response in the pest. Earlier, it was reported by Sinha and Krishna, 1971 [15] that an optimum level of terpenoid and a feeding habit widely distributed in cucurbitaceous plants was essential to stimulate feeding activity in *A. foveicollis*. At a lower concentration, these compounds will not initiate the beetle feeding, while higher concentrations act as a feeding deterrent. Thus, the differential feeding behavior of *A. foveicollis* could also be due to the varied concentrations of such secondary metabolites. However, field trials are equally important to test the responses of these identified compounds. Lastly, the Y tube olfactometric bioassay results under laboratory conditions clearly demonstrated the short-range olfactory responses of *A. foveicollis* to mostly long chain alkanes and a few other compounds, which are low volatile substances that might act as close range allelochemicals after the arrival of the chrysomelid beetle to the plant. Based on current findings from GC-MS of plant extract, EAG and Y-tube assay, 1, 4 dimethoxybenzene, heneicosane, decanal, pentacosane, heptadecane, octadecane, hexadecane, ethylbenzene and D-limonene are a few probable compounds that could be studied further for future formulation to facilitate the development of eco-friendly, baited traps as a strategy for insect pest management programs. However, to get more meaningful data, our future research focuses on identifying relevant odorants using single-sensillum recordings (SSR) from olfactory receptor neurons, followed by Gas chromatography–electroantennography (GC-EAG) of the emitted volatiles from the host plant, which would further help in providing detail and accurate insight into the neural mechanisms underlying these results, thereby leading to better designing of baited traps.

## 5. Conclusions

Based on the above study, the test insect pests showed more preference for saturated straight chain hydrocarbons/n-alkanes (C14, C17, C19, C20, C21 and C25), aromatic compound (1, 4 dimethoxybenzene) and green leaf volatile GLV (decanal) from the plant extract, which act as semiochemicals involved in host location.

**Supplementary Materials:** The following supporting information can be downloaded at: https://www.mdpi.com/article/10.3390/agronomy12112640/s1. Figure S1. Images of (A) Adult *Aulacophora foveicollis* Lucas (Coleoptera: Chrysomelidae). (B), (C) and (D) damaged leaves and flower of *Cucurbita maxima*. (E) Trench feeding behavior shown by the pest. Figure S2. Representative raw data of EAG spectra measured in mV of few stimulated odorant. Table S1. List of chemical compounds used for EAG. Table S2. Dose-dependent EAG amplitude (relative mean amplitude in %) of individual compound from different chemical classes in female *Aulacophora foveicollis*. Compounds in bold are present in the *C. maxima* volatile extract. Table S3. Dose-dependent EAG amplitude (relative mean amplitude in %) of individual compound from different chemical classes in male *Aulacophora foveicollis*. Compounds in bold are present in the *C. maxima* volatile extract Table S4. Representation of two-way ANOVA and post hoc tests analysis. Table S5. Y-tube assay [a] analysis in female and male *Aulacophora foveicollis*.

**Author Contributions:** Conceptualization, K.B.; Data curation, U.C. and B.P.; Formal analysis, B.B., U.C., A.M. and B.P.; Investigation, B.B.; Methodology, B.B., U.C., A.M. and B.P.; Project administration, K.B.; Resources, K.B.; Software, B.B. and U.C.; Supervision, K.B.; Validation, K.B.; Visualization, K.B.; Writing–original draft, K.B. All authors have read and agreed to the published version of the manuscript.

**Funding:** Corresponding author, KB, is grateful for financial support to Grants-in-Aid under RUSA Component 10 (IP/RUSA(C-10)/03/2021). The author is also grateful to the Department of Science & Technology-Uzbek (INT/UZBEK/P-09,2021) for the support. PRG, University of Kalyani & DST-PURSE 2022-23 are acknowledged for partial financial support.

**Data Availability Statement:** zenodo doi:10.5281/zenodo.7049100. File entitled "Raw data Female_final olf bioassay" and "Raw data male_final olf bioassay" demonstrates the datasheet regarding the Olfactometric Bioassay analysis of male and female *Aulacophora foveicollis*. File entitled "Raw data male female AF" showcases the datasheet for EAG experiment of *Aulacophora foveicollis*.

**Acknowledgments:** The authors are grateful to ICAR-Indian Agricultural Research Institute, Pusa Campus, New Delhi-110012 for providing the facilities for the EAG experiments and NCL Pune for providing the facility for SEM.

**Conflicts of Interest:** The authors declare that they have no conflict of interest.

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
