# Peer review of "Attraction of Aulacophora foveicollis Lucas (Coleoptera: Chrysomelidae) to Host Plant Cucurbita maxima Duchesne (Cucurbitaceae) Volatiles"

_agronomy, doi:10.3390/agronomy12112640_

Round 1
Reviewer 1 Report
These are my main comments on the manuscript (agronomy-1928291) entitled “Attraction of Aulacophora foveicollis Lucas to host plant Cucurbita maxima volatiles”. The manuscript investigates the olfactory responses of antennal chemosensilla by males and females A. foveicollis towards the plant volatiles were studied by electroantennogram. Following substantial revisions should be incorporated in the manuscript prior to acceptance.
1. I have concerns about the manuscript sections that I believe need to be addressed in order to improve its clarity.
2. A hypothesis for this work is needed.
3. In results section, some statistical analyses are missing.
4. Other revisions could be checked in PDF attached.

Author Response
Dear Sir,
We are extremely thankful to the Reviewer for his/her interest in our work and for critical and judicious evaluation of the manuscript, and many helpful comments that has greatly improved the manuscript. We have tried to do our best to respond to the points raised. As indicated below, we have checked all the general and specific comments provided by the reviewer and have made necessary changes (highlighted with tallow colour) according to the suggestion.
Title: “Attraction of Aulacophora foveicollis Lucas (Coleoptera: Chrysomelidae) to host plant Cucurbita maxima Duchesne (Family: Cucurbitaceae) volatiles”
Manuscript No. Agronomy-1928291
Answers to the points raised by the reviewer are as follows:
Report 1
These are my main comments on the manuscript (agronomy-1928291) entitled “Attraction of Aulacophora foveicollis Lucas to host plant Cucurbita maxima volatiles”. The manuscript investigates the olfactory responses of antennal chemosensilla by males and females A. foveicollis towards the plant volatiles were studied by electroantennogram. Following substantial revisions should be incorporated in the manuscript prior to acceptance.
- I have concerns about the manuscript sections that I believe need to be addressed in order to improve its clarity.
- A hypothesis for this work is needed.
- In results section, some statistical analyses are missing.
- Other revisions could be checked in PDF attached.
Answer: Tried to clarify the text, specially the introduction and conclusion part.
Hypothesis mentioned in the introduction is ‘This study specifies that semiochemicals involved in host location may contribute to novel and sustainable pest management program such as baited traps in future’ (last paragraph of the introduction).
Statistical analysis is mentioned in detail.
All corrections incorporated. Chysomelidae [36] family has been mentioned in general.

Reviewer 2 Report
Dear authors,
I read an interesting piece of science, however, I noticed several major concerns that need to be solved before its publication. At following I indicate this concerns. I recommend you to solve this concerns and the send your manuscript to a native speaker or an English editing service to revise English.
Good luck,
L2. Add Family and Order in parenthesis after the scientific name
L3 Add author, Family and Order in parenthesis after the scientific name
Abstract. There is no flow among the sentences. Please reword introductory sentence to be more related to the objective. Please do not use abbreviations without a mean (GC-MS).
L29-34. Please write an interpretation of your results instead using the symbols “>” or “=”
L40-41. Indicated what EAG, Y-TUBE, VOCs, GC-MS mean.
Introduction. Please re-structure introduction since in its present form is not clearly of which is the focus, model and system. Please consider addressee six paragraph in the following order: 1. Definition, biology and ecology of the red pumpkin beetle, 2. Economic importance, 3. Management strategies, 4. Causes of pest outbreaks (allelochemicals), 5. Similar recent studies., and 6. Definition of research problem, justification, objectives, hypothesis.
Materials and methods
L86. Add a map of the sites or fields where insects were collected.
L91. Please add a section named “Experimental design” where you state clearly the classical experimental design you followed, factors, levels of factors, response variables, experimental units, sampling units.
L197 – 203. Please state clearly in the section the reasons for log10 transforming data. I see that you used a 2-way ANOVA, so it is necessary to include at least three assumptions that your data presented.
L204 -205. Do not construct paragraph with only a single sentence. Please provide a cite for PAST. Please state the type and nature of the response variable and if it meets the assumptions of t-test.
Fig. 3. Indicate in Y-axis the name of the variable and its units of measure. Please do not use abbreviations in figure legend.
Fig. 4 Idem.
Fig. 5 Idem.
Fig. 6 Indicate in X-axis the name of the variable and its units of measure.
Conclusions. I noticed that there is no clear focus of this research since your first conclusion is not about the attraction of A. feveicollis to plant volatiles. Please remove results in this section. Please provide generalizations of your investigation.
L557-559 this is a speculative sentence. Please reword it or change this conclusion to something more realistic of your presented work.
Author Response
Report 2
Dear Sir,
We are extremely thankful to the Reviewer for his/her interest in our work and for critical and judicious evaluation of the manuscript, and many helpful comments that has greatly improved the manuscript. We have tried to do our best to respond to the points raised. As indicated below, we have checked all the general and specific comments provided by the reviewer and have made necessary changes (highlighted with tallow colour) according to the suggestion.
Title: “Attraction of Aulacophora foveicollis Lucas (Coleoptera: Chrysomelidae) to host plant Cucurbita maxima Duchesne (Family: Cucurbitaceae) volatiles”
Manuscript No. Agronomy-1928291
Answers to the points raised by the reviewer are as follows:
Dear authors,
I read an interesting piece of science, however, I noticed several major concerns that need to be solved before its publication. At following I indicate this concerns. I recommend you to solve this concerns and the send your manuscript to a native speaker or an English editing service to revise English.
Good luck,
L2. Add Family and Order in parenthesis after the scientific name.
Answer: Incorporated
L3 Add author, Family and Order in parenthesis after the scientific name.
Answer: Incorporated
Abstract. There is no flow among the sentences. Please reword introductory sentence to be more related to the objective. Please do not use abbreviations without a mean (GC-MS).
Answer: Corrections incorporated.
L29-34. Please write an interpretation of your results instead using the symbols “>” or “=”
Answer: Corrected in the abstract part (line 29-34).
L40-41. Indicated what EAG, Y-TUBE, VOCs, GC-MS mean.
Answer: All mentioned
Introduction. Please re-structure introduction since in its present form is not clearly of which is the focus, model and system. Please consider addressee six paragraph in the following order: 1. Definition, biology and ecology of the red pumpkin beetle, 2. Economic importance, 3. Management strategies, 4. Causes of pest outbreaks (allelochemicals), 5. Similar recent studies., and 6. Definition of research problem, justification, objectives, hypothesis.
Answer: Introduction has been paragraphed as per suggestion.
Materials and methods
L86. Add a map of the sites or fields where insects were collected.
Answer: Location with latitude and longitude is mentioned.
L91. Please add a section named “Experimental design” where you state clearly the classical experimental design you followed, factors, levels of factors, response variables, experimental units, sampling units.
Answer: Experimental design- For dose-response studies, the experiments were laid out in completely randomized design with four treatments and four replications. Twenty 24-48hrs old insects were considered for each replication. For Y tube bioassay also, the experiment was laid out in completely randomized design format with five replications comprising of twenty insects in each replication. The two-way ANOVA compares the mean differences between groups that have been split on two independent variables (called factors). The primary purpose of a two-way ANOVA is to understand if there is an interaction between the two independent variables on the dependent variable. Here our independent variables are female insects, male insects and concentrations and the dependent variable is response of the insects.
L197 – 203. Please state clearly in the section the reasons for log10 transforming data. I see that you used a 2-way ANOVA, so it is necessary to include at least three assumptions that your data presented.
Answer: Log10 transformation was a typographical error. We are extremely sorry for this sort of mistake.
L204 -205. Do not construct paragraph with only a single sentence. Please provide a cite for PAST. Please state the type and nature of the response variable and if it meets the assumptions of t-test.
Answer: Reference- Hammer Ø, Harper DAT, Ryan PD (2001) PAST: paleontological statistics software package for education and data analysis. Palaeontologia. Electronica 4(1): 9; http:// palaeo- elect ronica. org/ 2001_1/past/ issue1_ 01. html.
t-test was performed, to study the variation in the distribution of sensilla on circumferential band of male and female; furthermore, to analyse the result of olfactory bioassay or Y-tube assay, t-test was also performed within control and treatment groups. Before performing t-test, all the data was tested for homogeneity of variance by using Levene’s test, where, data with non-significant Levene’s test values considered as homogenous (equal variance) and subjected to t-test for equal variance.
In the manuscript, methodology section for t-test analysis was rewritten with the above-mentioned information. For reviewer, two tables are being provided below regarding the Levene’s test (Table XA and XB).
|
Table XA: Representation of p-values of Levene’s test for analysing the homogeneity of variance between treatment and control group in olfactory bioassay. A total number of five replicates with 20 individuals in each have been studied in this assay i.e., N= 100. |
|
|
Olfactory Bioassay |
|
|
Compounds |
Levene’s test p-value |
|
Male |
|
|
(1S)-(-)-α-pinene |
0.76 |
|
Decanal |
0.15 |
|
Phytol |
0.24 |
|
Nonanal |
0.58 |
|
1,4 dimethoxybenzene |
0.22 |
|
Heneicosane |
1 |
|
Tetradecane |
0.07 |
|
Pentacosane |
0.44 |
|
Ethylbenzene |
0.46 |
|
Benzaldehyde |
0.24 |
|
D-limonene |
0.76 |
|
Nonadecane |
0.39 |
|
Eicosane |
0.75 |
|
Female |
|
|
(1S)-(-)-α-pinene |
0.21 |
|
Decanal |
1 |
|
Phytol |
1 |
|
Nonanal |
0.44 |
|
1,4 dimethoxybenzene |
0.31 |
|
Heneicosane |
1 |
|
Tetradecane |
0.75 |
|
Pentacosane |
1 |
|
Ethylbenzene |
0.46 |
|
Benzaldehyde |
0.24 |
|
D-limonene |
0.21 |
|
Nonadecane |
0.24 |
|
Eicosane |
0.21 |
|
Table XB: Representation of p-values of Levene’s test for analysing the homogeneity of variance between male (N= 10) and female (N=10) regarding sensilla distribution of circumferential band. |
|
|
Sensilla distribution |
|
|
Sensilla type |
Levene’s test p-value |
|
Dorsal |
|
|
SB-I |
0.91 |
|
SB-II |
0.55 |
|
SB-III |
0.07 |
|
SB-IV |
0.14 |
|
SB-V |
0.36 |
|
SB-VI |
0.21 |
|
SB-VII |
0.36 |
|
SCH-I |
0.74 |
|
SCH-II |
0.46 |
|
ST |
0.44 |
|
Ventral |
|
|
SB-I |
0.52 |
|
SB-II |
0.65 |
|
SB-III |
0.33 |
|
SB-IV |
0.77 |
|
SB-V |
0.82 |
|
SB-VI |
0.23 |
|
SB-VII |
0.72 |
|
SCH-I |
0.15 |
|
SCH-II |
0.48 |
|
ST |
0.26 |
|
SB-I |
0.21 |
|
SB-II |
0.24 |
|
SB-III |
0.21 |
Fig. 3. Indicate in Y-axis the name of the variable and its units of measure. Please do not use abbreviations in figure legend.
Answer: Corrected
Fig. 4 Idem.
Answer: Corrected
Fig. 5 Idem.
Answer: Corrected
Fig. 6 Indicate in X-axis the name of the variable and its units of measure.
Answer: Fig. 6 clarified
Conclusions. I noticed that there is no clear focus of this research since your first conclusion is not about the attraction of A. feveicollis to plant volatiles. Please remove results in this section. Please provide generalizations of your investigation.
L557-559 this is a speculative sentence. Please reword it or change this conclusion to something more realistic of your presented work.
Answer: Conclusion is rewritten and briefed.

Reviewer 3 Report
Dear authors,
I believe your research is very interesting and useful. It needs to be better presented however. Pay attention to the formulation and grammar as many messages are not clear. Review your method and results. Please see the few points below that are particularly important.
Method: please clarify the methods
l113 you washed the porapaq only with hexane? Do you mean you collected in hexane, the headspace from porapaq...?
did you do a control collection? how did you clean the porapaq to ensure no contaminations between samples?
l143 how did you identify the compounds? the current explanation is not a correct method. Have you used standards? use Kovat indexes? co-elution?
gc-eag would have been much better to identify the antennally active compounds, instead of litterature.
1146 why 2day-old?
l148 so 20 males and 20 females= 40 insects? or 40 males and 40 females so 80 insects?
l166. what do you mean by standard? is it a positive control? negative control? to control the responsiveness of the antenna?
I do not understand the purpose of the EAG data calculations, and why the log10? and how did it help? what did you calculate, why, how?
l196 please review, it is very unclear. be very specific on the reason why doing the calculations and tests. explain them well.
results
table 1. why are two lines highlighted? is the list of alcanes found inside the extract? why don't you make a figure or table highlighting the differences between intact and damaged plants?
table 2and figure 3/4 difficult to read. try changing it or make a figure which show clearly the male/female differences. it must be easy to read.
perhaps make a figure which include both the sensilla distribution and the eag responses? to link the differences and their implications!
the dose response should be seen as a dose-response curve to really appreciate what it means.
male/female should appear in parallel, not in separate graphs, if that is what you are comparing.
as you also demark the different chemical classes, perhaps the color code or graph panel could be arranged per chemical class? this would bring more clarity: for each chemical class, show the dose/response curve to each chemical by male and female. show the statistical test which compare male and female!
The statistical analysis is not a special result section. it is part of the result. focus on one question you ask, then describe the result you obtain.
l445 based on.... is not clear. specify why they were chosen.
figure 6 very bad quality and as for other figures: arrange it so it shows the results you want to show.
Best of luck
Author Response
Dear Sir,
We are extremely thankful to the Reviewer for his/her interest in our work and for critical and judicious evaluation of the manuscript, and many helpful comments that has greatly improved the manuscript. We have tried to do our best to respond to the points raised. As indicated below, we have checked all the general and specific comments provided by the reviewer and have made necessary changes (highlighted with tallow colour) according to the suggestion.
Title: “Attraction of Aulacophora foveicollis Lucas (Coleoptera: Chrysomelidae) to host plant Cucurbita maxima Duchesne (Family: Cucurbitaceae) volatiles”
Manuscript No. Agronomy-1928291
Answers to the points raised by the reviewer are as follows:
Dear authors,
I believe your research is very interesting and useful. It needs to be better presented however. Pay attention to the formulation and grammar as many messages are not clear. Review your method and results. Please see the few points below that are particularly important.
Method: please clarify the methods
l113 you washed the porapaq only with hexane? Do you mean you collected in hexane, the headspace from porapaq...?
Answer: Yes
did you do a control collection? how did you clean the porapaq to ensure no contaminations between samples?
Answer: Porapaq is cleaned and made contamination free by washing with diethyl ether (60 mL), methanol (80 mL), and deionized water (100 mL).
l143 how did you identify the compounds? the current explanation is not a correct method. Have you used standards? use Kovat indexes? co-elution?
Answer: The mass spectra were analysed and identified using software Turbo-Mass-OCPTVS-Demo SPL, GC-MS library NIST14 and other synthetic compounds were directly procured from commercial suppliers. Kovat indexes was not used. For GC-MS hexane was used as standard.
gc-eag would have been much better to identify the antennally active compounds, instead of litterature.
Answer: We completely agree with the reviewer and in future we shall focus on GC-EAG technique.
1146 why 2day-old?
Answer: 24-48 hr old insects are in the peak of their vigourous health.
l148 so 20 males and 20 females= 40 insects? or 40 males and 40 females so 80 insects?
Answer: 40 females and 40 males.
l166. what do you mean by standard? is it a positive control? negative control? to control the responsiveness of the antenna?
Answer: 1-hexanol is the standard taken, it is the positive control
I do not understand the purpose of the EAG data calculations, and why the log10? and how did it help? what did you calculate, why, how?l196 please review, it is very unclear. be very specific on the reason why doing the calculations and tests. explain them well.
Answer: EAG values (mV) were corrected by subtracting it from the values of paraffin oil (used for diluting and making various concentrations) obtained. The data were then standardized by expressing the corrected mean EAG values (mV) as a percentage of the standard stimulus. The data recorded in percentage was then subjected to arcsine transformations. The arcsine transformation used is a standard procedure to make highly skewed distributions less skewed. This is valuable both for making patterns in the data meet the assumptions of inferential statistics. These arcsines transformed relative values were subjected to 2-way ANOVA using software SPSS 16.0. Subsequently, within each sex, a one-way ANOVA was computed. After ANOVA, post hoc analysis was performed to explore the mean differences between pairs of groups. The contrasts between chemicals were examined by the Scheffe’s contrast method. t-test analysis was performed using PAST software version 4.05 to understand the variation in the distribution of sensilla on the circumferential band of the insect pest.
results
table 1. why are two lines highlighted? is the list of alcanes found inside the extract? why don't you make a figure or table highlighting the differences between intact and damaged plants?
Answer: Peak no. 10 with the highest peak area and peak no.15 as the lowest peak area hence highlighted, and areas peak below this (peak no. 15) were not considered.
In Figure 1 A & B, we have already showed the comparative presentation of the two types of volatile compounds extracted from the undamaged and conspecific damaged plant. The difference lies in the peak area of the compounds (which is shown very clearly in the figure). Since no new compound was found in case of consfecific damaged plant so the damaged compound list was not shown again in the table 1.
table 2and figure 3/4 difficult to read. try changing it or make a figure which show clearly the male/female differences. it must be easy to read.
Answer: Table and figures both were clarified as per suggestion.
perhaps make a figure which include both the sensilla distribution and the eag responses? to link the differences and their implications!
Answer: Unless we do the single sensillum recording analysis, it is not suggested to present it together with the EAG data.
the dose response should be seen as a dose-response curve to really appreciate what it means.
Answer: We have calculated the dose response curve based on R2 value (see section 3.5 Result part) for all 50 volatile compounds in female and male. In the text we have mentioned the trend of the R2 values and no individual values have been shown. Accordingly, we have categorised the compounds as most responsive compound, moderately responsive compounds and erratic responsive compounds but if reviewer wants it can be presented in the supplementary file (then the file will become too big, please suggest).
male/female should appear in parallel, not in separate graphs, if that is what you are comparing.
as you also demark the different chemical classes, perhaps the color code or graph panel could be arranged per chemical class? this would bring more clarity: for each chemical class, show the dose/response curve to each chemical by male and female. show the statistical test which compare male and female!
Answer: Figure 3, 4 & 5 (re drawn with high resolution) have been presented to compare the EAG responses in female and male based on compounds (chemical classes), doses and relative mean amplitudes. Statistical analysis with error bar is also shown in the figures and given in Table S2 & S3.
Still, we are open to the decision of reviewer, whether the figures need to be presented in any other way or not. For convenience we have shown a representative figure (below mean EAG data at 5 mg/ml in female and male), but we think it looks clumsier.
The statistical analysis is not a special result section. it is part of the result. focus on one question you ask, then describe the result you obtain.
Answer: Statistical analysis is a part of Materials and methods.
l445 based on.... is not clear. specify why they were chosen.
Answer: Based on the headspace volatile extracts from C. maxima which were obtained in maximum to moderate amount (based on peak area) and in EAG responses of the compounds that showed high to moderate stimulation, compounds were selected for the Y-tube olfactometric bioassay.
figure 6 very bad quality and as for other figures: arrange it so it shows the results you want to show.
Answer: Fig 6 has been clarified.
Best of luck

Round 2
Reviewer 1 Report
The authors have incorporated all suggestions and comments into the revised version, now the manuscript seems much clear. There is minor point to be corrected:
Ls.68-59: …to A. foveicollis…
L.114: absolute alcohol is 99% and not 100%. Correct
L.160: The acronym EAG correspond to “Electroantennography”. Please, correct in all manuscript.
L.189: … attractiveness in A. foveicollis adults…
L.230: … analysis of C. maxima plant…
L.279: Delete “,” before 18
Ls.283-288: In figure 2, four images (see upper plate) about female/male antennae were not mentioned and detailed in figure legend.
Table 2: In ventral side (SB-I x P-value), delete “P<”. Also, letter “P” should be in italic and minuscule.
Ls.351-352: …of A. foveicollis…
Ls.536-537: … data in females and males of A. foveicollis…
Ls.548-551: Revise this sentence to eliminate rewordiness
Ls.583-584: … emitted from C. maxima…
Ls.594: Momordica charantia
Ls.636-637: For these insect species, provide the ID author, family and order taxa.
Author Response
Report 1
The authors have incorporated all suggestions and comments into the revised version, now the manuscript seems much clear. There is minor point to be corrected:
Ls.68-59: …to A. foveicollis…
Answer: corrected
L.114: absolute alcohol is 99% and not 100%. Correct
Answer: corrected
L.160: The acronym EAG correspond to “Electroantennography”. Please, correct in all manuscript.
Answer: corrected.
L.189: … attractiveness in A. foveicollis adults…
Answer: corrected
L.230: … analysis of C. maxima plant…
Answer: corrected
L.279: Delete “,” before 18
Answer: corrected
Ls.283-288: In figure 2, four images (see upper plate) about female/male antennae were not mentioned and detailed in figure legend.
Answer: corrected
Table 2: In ventral side (SB-I x P-value), delete “P<”. Also, letter “P” should be in italic and minuscule.
Answer: corrected
Ls.351-352: …of A. foveicollis…
Ls.536-537: … data in females and males of A. foveicollis…
Ls.548-551: Revise this sentence to eliminate rewordiness
Ls.583-584: … emitted from C. maxima…
Ls.594: Momordica charantia (It should be Momordica cochinchinensis)
Answer: All corrections incorporated.
Ls.636-637: For these insect species, provide the ID author, family and order taxa.
Answer: corrections incorporated.

Reviewer 2 Report
Dear authors,
in the attachment you can find some suggestions.
good luck

Author Response
Report 2
Comments and Suggestions for Authors
Dear authors,
in the attachment you can find some suggestions.
good luck
Answer: All corrections incorporated according to the suggestion and highlighted with blue colour.

Reviewer 3 Report
Dear author,
this is much better, please review for typos and grammar.
Best,
Author Response
Report 3
Comments and Suggestions for Authors
Dear author,
this is much better, please review for typos and grammar.
Best,
Answer: Typographical errors taken care
